# Development of a Probabilistic Seismic Performance Assessment Model of Slope Using Machine Learning Methods

**Shinyoung Kwag [1], Daegi Hahm [2], Minkyu Kim [2] and Seunghyun Eem [3,*]** 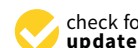

[1]  Department of Civil and Environmental Engineering, Hanbat National University, Daejeon 34158, Korea; skwag@hanbat.ac.kr

[2]  Mechanical and Structural Safety Research Division, Korea Atomic Energy Research Institute, Daejeon 34057, Korea; dhahm@kaeri.re.kr (D.H.); minkyu@kaeri.re.kr (M.K.)

[3]  School of Convergence & Fusion System Engineering, Kyungpook National University, Gyeongsanbuk-do 37224, Korea

*  Correspondence: eemsh@knu.ac.kr; Tel.: +82-54-530-1483

**Abstract:** The objective of this study is to propose a model that can predict the seismic performance of slope relatively accurately and efficiently by using machine learning methods. Probabilistic seismic fragility analyses of the slope had been carried out in other studies, and a closed-form equation for slope seismic performance was proposed through a multiple linear regression analysis. However, the traditional statistical linear regression analysis showed a limit that could not accurately represent such nonlinear slope seismic performances. To overcome this limit, in this study, we used three machine learning methods (i.e., support vector machine (SVM), artificial neural network (ANN), Gaussian process regression (GPR)) to generate prediction models of the slope seismic performance. The models obtained through the machine learning methods basically showed better performance compared to the models of the traditional statistical methods. The results of the SVM showed no significant performance difference compared with the results of the nonlinear regression analysis method, but the results based on the ANN and GPR showed a remarkable improvement in the prediction performance over the other models. Furthermore, this study confirmed that the GPR-based model predicted relatively accurate seismic performance values compared with the model through the ANN.

**Keywords:** slope seismic performance; machine learning methods; support vector machine (SVM); artificial neural network (ANN); Gaussian process regression (GPR)

## 1. Introduction

The destruction of slopes caused by earthquakes can directly or indirectly cause serious damage to the neighboring society. Specifically, these earthquake-induced slope failures can directly cause serious damage to various major facilities, such as nearby buildings, roads, and the plant, and can result in human casualties [1]. Additionally, damage to hospitals, roads, and major plants may paralyze social functioning, which could lead to severe economic losses and extensive indirect casualties. Earthquake-induced slope collapses occurred in the 1970 Peru earthquake, the 1990 Luzon earthquake in the Philippines, the 1999 Chi-Chi earthquake in Taiwan, the 2001 El Salvador earthquake, the 2008 Wenchuan earthquake in China, the 2011 Christchurch earthquake in New Zealand, the 2016 Kumamoto earthquake in Japan, etc., and much damage was reported in each case [2]. Large earthquakes generally do not occur in Korea, and accordingly, there has been no significant damage caused by such earthquakes. However, since the start of monitoring and measuring earthquake motions in 1978,

earthquakes in Korea with a moment magnitude of 2.0 or higher have occurred 20 times per year until 1998, but this frequency has increased to an average of 48 times per year during the period between 1998 and 2015 [3]. Furthermore, Korea has experienced two strong earthquakes, one in Gyeongju in 2016 and the other in Pohang in 2017, raising concerns over safety during earthquakes near major facilities [1,4–6].

There are three primary methods of assessing the stability of slopes during earthquakes, which are categorized as a pseudostatic analysis, Newmark permanent displacement analysis, and stress-deformation analysis. The pseudostatic analysis method [7] assumes statically the seismic load, thereby deriving safety factors through considering the static equilibrium state at the collapse planar of the slope, and evaluating the stability of the slope based on these results. The Newmark permanent displacement analysis method [8] assumes that the soil medium above a failure planar of the slope is a rigid block, and draws permanent displacement based on the dynamic motions of the rigid block under dynamic earthquake loadings. The permanent displacement thus obtained is utilized as an objective indicator to assess the stability of the slope caused by earthquakes. The stress-deformation analysis method evaluates the seismic stability of slopes through a dynamic numerical analysis based on numerical analysis methods: the finite element method, the finite difference method, the discontinuous deformation analysis, smoothed particle hydrodynamics, etc. Each of the three primary methods has advantages and disadvantages and are chosen and applied according to circumstances. In general, the c-static analysis is simple to apply, but the accuracy is lowered due to an excessive assumption. On the other hand, the stress-deformation analysis method through a numerical model improves accuracy in a detailed analysis. However, the drawback is that it demands values on many parameters that need to be determined and it costs considerable computation time. Therefore, the Newmark permanent displacement analysis method has been frequently used by many researchers as an appropriate compromise between the other methods [9]. However, there is a disadvantage for this method in that the analysis time is longer than that of the pseudostatic analysis method, as it performs a time–history analysis based on a particular earthquake recording. However, to overcome this issue, diverse empirical equations, which calculate a permanent displacement without doing dynamic time–history analyses on the specific earthquake loading history, had been proposed in other studies [9–16].

Previous studies [17–24] attempted to develop the method of probabilistic evaluation of the stability of a slope considering the randomness and uncertainties of the slope model and earthquake ground motions. The randomness and uncertainties are due to various causes, such as measurement error, modeling method, various assumptions, etc., and thus, to show the variabilities of values due to such uncertainties, the soil parameters for the slope are assumed to be probability distribution functions with a specific coefficient of variations. For information on specific coefficients of variation, guidelines have been provided from the statistical data on soil [17]. Specifically, when looking at probabilistic studies evaluating slope stability due to earthquakes, they have usually been conducted in the form of probabilistic seismic fragility assessments of the slopes. Yiannis et al. [18] and Wu [19] evaluated the stability of the slope for earthquakes based on the pseudostatic analysis, and they ultimately computed the slope seismic fragilities concerning the related randomness and uncertainties using the Monte Carlo simulation and the First-Order Reliability Method, respectively. Xu and Low [20], Huang et al. [21], Kim and Sitar [22], and others assessed the slope vulnerability based on the concept of the Newmark permanent displacement concept or dynamic numerical analysis method. However, the probabilistic analysis, which considers the uncertainties and randomness of the slope soil and earthquakes, has a clear limitation: It requires a large computation cost. Hence, to make up for this drawback, there have been studies to develop an efficient seismic vulnerability analysis method using an Artificial Neural Network (ANN) based on a pseudostatic analysis technique [23,24].

The slope stability evaluation through the seismic fragility analysis is a relatively complete and accurate method of providing slope failure probabilities over all seismic intensity regions. Besides, that analysis technique evaluates the seismic performance of the slope, not by using

an indirect indicator such as slope permanent displacement due to an earthquake, but by utilizing an intuitive index, such as peak ground acceleration, which denotes the direct intensity of the earthquake. However, the calculation cost for performing the seismic fragility analysis is excessive, and accordingly, it is difficult to apply the advantages of such analysis to an efficient seismic performance evaluation under given slope conditions. Moreover, its shortcoming appears evident when using this analysis for the development of an earthquake-induced slope vulnerability map. This is because many slopes of various geomorphological and geophysical conditions are spatially distributed, and thus, this requires a large computation amount for their seismic performance evaluation. Accordingly, Kwag and Hahm [25] proposed an efficient analysis method on evaluating a slope seismic fragility based on the concept of Newmark permanent displacement analysis. Moreover, also considering this method, they developed a linear regression model that could evaluate the seismic performance of the slope [26]. However, the developed linear regression model was basically one that was made by a method using conventional statistical multiple linear regression and had the disadvantage of producing a relatively large difference in accuracy compared with the original data. In other words, the traditional linear regression analysis method showed a limitation that could not accurately predict the nonlinear relationship between the slope of various conditions and the seismic performance. Such a nonlinear relationship was clearly evident in the original data. Therefore, in order to derive an accurate seismic performance prediction model of the slope, it is judged that it is essential to introduce a new method that can well simulate the nonlinear relationship of the original data.

Therefore, the purpose of this study is to focus on improving the accuracy of the previous model (which was developed by a linear regression analysis) by utilizing machine learning methods. Thus, the primary contribution of this study lies in the development of the more accurate model which can well predict the slope seismic performance compared to the previous model. To this end, based on the probabilistic seismic performance data for the slopes obtained from various slope conditions which were presented in the previous study, three machine learning methods such as an artificial neural network (ANN), a support vector machine (SVM) and Gaussian process regression (GPR) were adopted in this study. The reason for the use of these three machine learning methods in this study is that these methods are revealed to be well suited for predicting geotechnical engineering data [27–29]. Also, this lays emphasis on finding a more accurate prediction model by applying various learning algorithms to the same data and checking the result difference between individual learning algorithms. In this study, the ensemble learning method [30] or the hybrid machine learning method [31], which uses or combines multiple learning algorithms to obtain good prediction performance, were not applied for these data. This is because this study is focused on developing predictive models based on individual learning algorithms. The study on the application of more advanced machine learning methods is likely to be desirable to proceed as future research after identifying the results of individual algorithm application in this study.

## 2. Materials and Methods

### 2.1. Data Acquisition: Seismic Slope Fragility Assessment

In the theory of the Newmark permanent displacement analysis [8], the soil stratum above the failure planar of the slope is regarded as a single rigid block as shown in Figure 1a, and no internal deformation occurs inside the rigid block. Specifically, in Figure 1a, the red shaded area of the slope is treated as a sliding block, and the sloping dotted black line of the slope indicates the inclined plane which is frictionally supporting the sliding block. Under these assumptions, the acceleration at which the rigid block begins to move at a particular earthquake acceleration loading entering the ground is called the yield acceleration. If the earthquake acceleration recording exceeds the yield acceleration, the rigid block begins to move and then the permanent displacement is accumulated until the earthquake acceleration recordings are the same as the yield acceleration. In other words,

the permanent displacement of the slope can be finally estimated by calculating the accumulated value through the double integration of the difference between the earthquake acceleration recordings and the yield acceleration, as shown in Figure 1b.

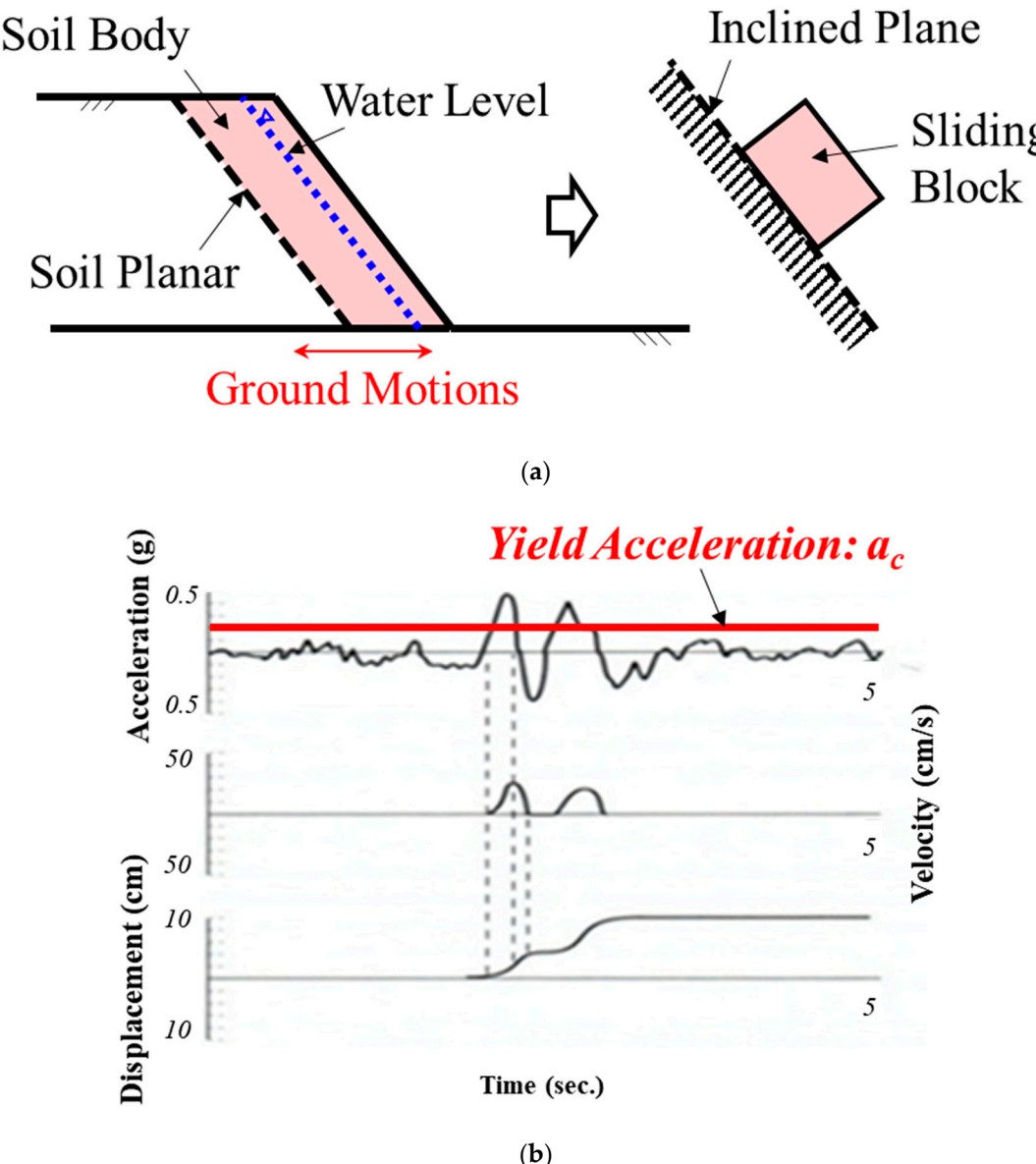

(a)

(b)

**Figure 1.** Concept of Newmark permanent displacement analysis: (**a**) Concept; (**b**) Calculation.

The Newmark permanent displacement analysis method has a disadvantage in that it cannot account for the amplification effect and damping effect of input seismic waves due to the deformation characteristics in the block, because it treats the moving block as a rigid body. Furthermore, since the strength change of the ground due to the vibration cannot be considered, this approach has a limitation in that the yield acceleration is assumed as a constant value while the analyses are being performed [32]. Nevertheless, that analysis through lab tests [33] and real cases [34] is known to predict relatively well the permanent displacement of slopes during earthquakes. In this context, other studies proposed empirical equations for predicting permanent displacements using this method, based on measured seismic data [10–16]. This came to be an opportunity to overcome the disadvantage of the Newmark permanent displacement analysis method, in which the analysis time is long because of the need to perform a time–history analysis based on the specific earthquake records as compared with the pseudostatic analysis method. The following equation shows

the empirical formula of the permanent displacement, proposed by Jibson [15], which we utilized in this study.

$$\log(D_n) = 0.215 + \log\left[\left(1 - \frac{a_c}{a_{\max}}\right)^{2.341}\left(\frac{a_c}{a_{\max}}\right)^{-1.438}\right] \pm 0.510 \tag{1}$$

$$a_c = (FS - 1) \cdot \sin(\alpha) \tag{2}$$

$$FS = \frac{c}{\gamma \cdot t \cdot \sin(\alpha)} + \frac{\tan(\varphi)}{\tan(\alpha)} - \frac{\gamma_w \cdot m \cdot \tan(\varphi)}{\gamma \cdot \tan(\alpha)} \tag{3}$$

where $D_n$ is the permanent displacement (cm), $a_{max}$ is the peak ground acceleration, and $a_c$ is the yield acceleration. The yield acceleration can be expressed as a function of the static safety factor ($FS$) and slope angle ($\alpha$, degree), as shown in Equation (2). The static safety factor is calculated through effective cohesion ($c$), soil unit weight ($\gamma$), internal friction angle ($\varphi$), water unit weight ($\gamma_w$), soil normal thickness of the failure surface ($t$), and the saturation ratio of the failure soil thickness ($m$), as shown in Equation (3). The right-hand side term ±0.510 in Equation (1) represents the standard deviation for the model. This model equation was derived from data obtained from the Newmark permanent displacement analyses based on a collection of 2270 strong-motion records observed in 30 worldwide earthquakes around the world. This expression has an $R^2$ value of 84%, which ensures statistical significance for the data. More information on this model is well described in Jibson [15].

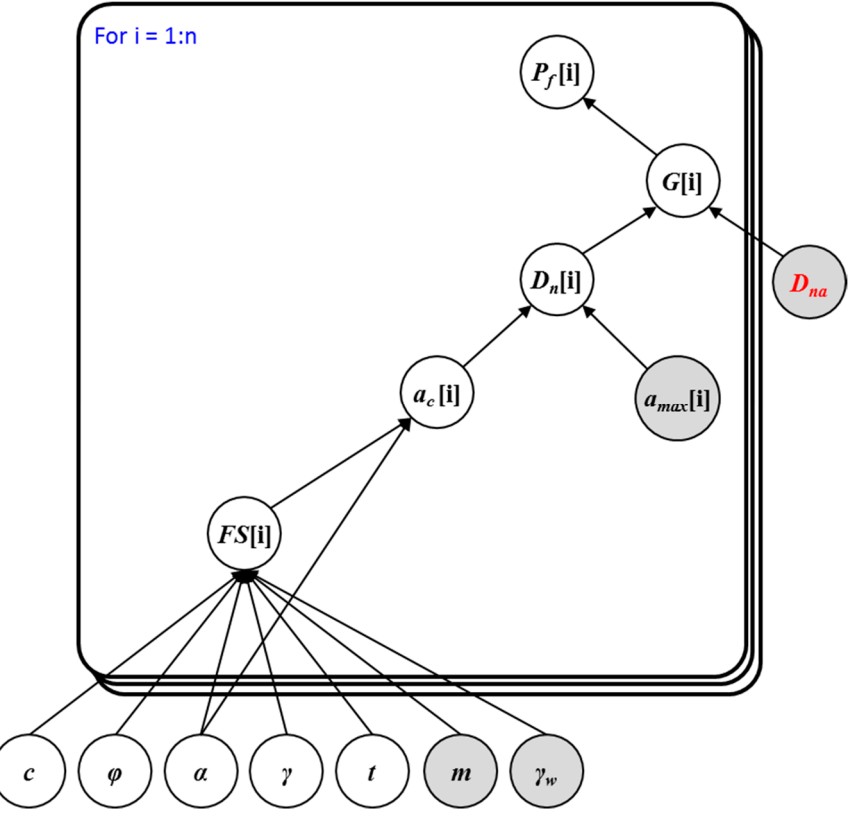

**Figure 2.** Conceptual process of earthquake-induced slope fragility evaluation [25,26]: The white circle denotes the random variables and the shaded circle represents the deterministic variables. A circle having $D_{na}$ indicates the limit state of permanent displacement which causes a slope seismic failure.

Earthquake-induced slope fragility analysis, which should be a probabilistic approach, requires analyses of diverse conditions to account for the randomness of the earthquake and the uncertainty of the slope model. Also, it is essential to ensure credibility or confidence in the criteria that define the slope failure. Accordingly, Jibson et al. [14] analyzed the correlation between predicted Newmark permanent displacements and slope failures observed during the Northridge

earthquake, based on the slopes of various ground conditions observed in the Oat Mountain Quadrangle (CA, USA) region. Kwag and Hahm [25,26] derived the critical permanent displacements causing the earthquake-induced slope failure by utilizing the traditional statistical methods, based on a correlation study on the permanent displacement-observed slope failure. To this end, a method of evaluating the earthquake-induced slope vulnerability (see Figure 2) was also proposed using the Monte Carlo sampling technique, with a basis on the empirical equation of Newmark permanent displacement [25]. That study once again proved the effectiveness of the study which qualitatively presented that the existing Newmark permanent displacement of 2 to 20 cm is a criterion for the risk of slope failure due to an earthquake [35]. Finally, the concept of the high-confidence-low-probability-of-failure (HCLPF) was employed to describe the seismic fragilities of the various slopes as single values of seismic performances considering the randomness and the uncertainties of slopes. Accordingly, the probabilistic seismic performance of the slope (i.e., HCLPF) with various conditions was derived [26].

Specifically, a seismic fragility analysis of a particular slope derives various fragility curves due to randomness and uncertainty. If we statistically deal with these fragility curves and summarize, these can be expressed as a mean fragility curve, a 5%-confidence-level fragility curve, and a 95%-confidence-level fragility curve (see Figure 3). These distribution expressions can represent the comprehensive seismic performance of the slope model, but it cannot be expressed as a representative single value. Therefore, that study introduced the HCLPF concept. This concept mathematically represents a 5% failure probability in the 95%-confidence-level fragility curve [36], and can be a measure of probabilistic seismic performance reflecting the uncertainty of the fragility curves and summarizing these results [37]. That is, using this concept, a single seismic performance value of the slope can be obtained considering the randomness and uncertainty of the earthquake and slope model. Owing to such an HCLPF feature, this concept has been widely used to represent the probabilistic seismic performance of major structures, systems, and components in a nuclear facility as a representative value.

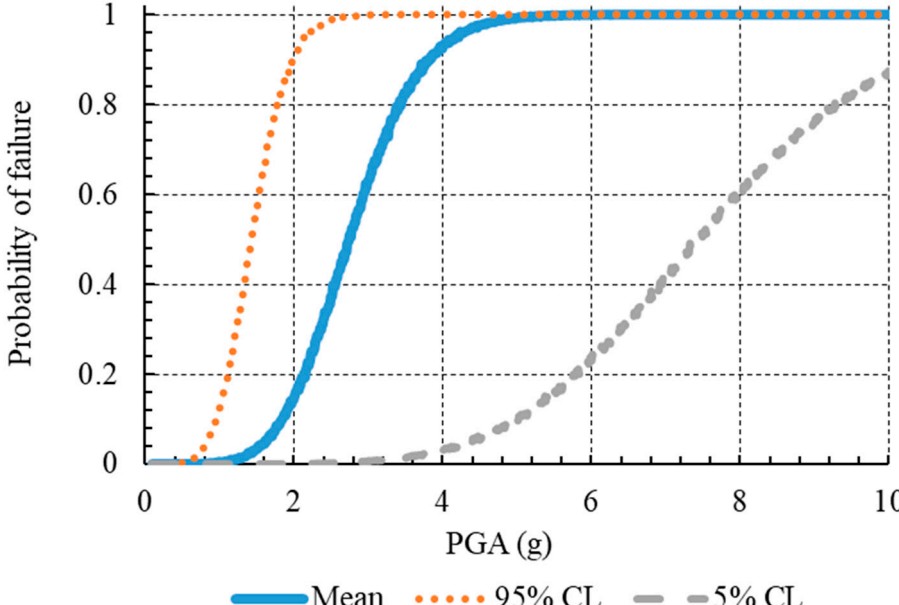

**Figure 3.** Seismic fragility curves for a particular slope.

The study examined the effect of the change of input variables on the slope HCLPF seismic performance. The HCLPF values were evaluated according to each change of the input variable value, with a basis on the slope condition of Figure 3 ($c$ = 50 kPa, $\varphi$ = 30°, $\alpha$ = 45°, $\gamma$ = 16 kN/m$^3$, $t$ = 3 m, $m$ = 0, $\gamma_w$ = 9.807 kN/m$^3$), and Figure 4 represents the results. Figure 4 shows that the HCLPF seismic performance increases with increasing soil cohesion and internal friction angle. On the other

hand, Figure 4 shows that the HCLPF seismic performance decreases as the slope angle, the soil unit weight, the normal thickness of the slope failure planar, and the saturation ratio increase. Based on this tendency, the additional analysis was performed to examine the HCLPF seismic performance change with the arbitrary change of all variables in the multidimensional input variable space. Such additional analysis results introduced in the next subsection denote the raw data that we utilized for this study.

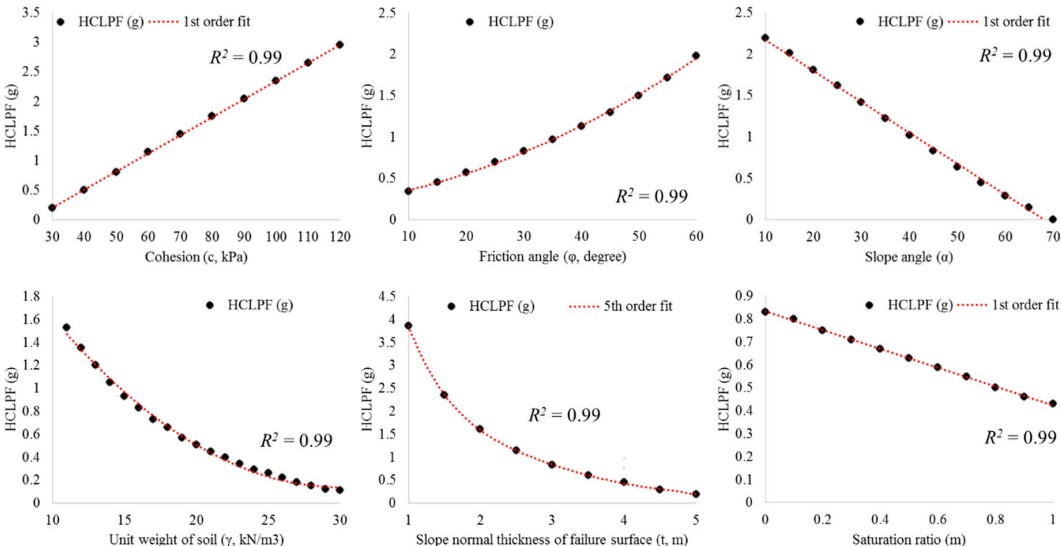

**Figure 4.** HCLPF seismic performances of slope according to changes in each slope model properties of Figure 3. HCLPF: high-confidence-low-probability-of-failure.

### 2.2. Data Description

Various slope conditions of the original data were made by the following six major factors that determine the Newmark permanent displacement: $c$ = 30–120 kPa; $\varphi$ = 10–60°; $\alpha$ = 10–70°; $\gamma$ = 11–30 kN/m$^3$; $t$ = 1–5 m; and $m$ = 0–1. The 10,000 various slope conditions were extracted through the Latin hypercube sampling technique [38] to efficiently consider all possible combinations in multidimensional space composed of six parameters. Table 1 summarizes the range and distribution of values for these major input and output parameters. Based on the 10,000 slope conditions thus extracted, the HCLPF seismic performances of the slopes were evaluated through analyses of slope vulnerabilities subjected to earthquakes and were databased. The permanent displacement for the input variables in the slope seismic fragility analysis was calculated using Equations (1)–(3). The limit state of permanent displacement ($D_{na}$) causing slope seismic failure was used as the permanent displacement value obtained based on actual observation data. Finally, the probability of slope failure according to the earthquake intensity was performed through the Monte Carlo simulation method. Figure 2 schematically shows the relationship between input variables, permanent displacement, limit permanent displacement, and slope failure probability as a method of analyzing the seismic fragility of slopes. Here, HCLPF was defined as the value of a 5% failure probability in the 95% confidence interval of the derived slope seismic fragility curve. The previous study [25,26] presented a multiple linear regression (MLR) model by statistically analyzing these original data (see Figure 5) as follows:

$$
\begin{aligned}
HCLPF = \quad & 5.14 + 2.94{\cdot}10^{-5}{\cdot}c + 2.43{\cdot}10^{-2}{\cdot}\varphi - 3.48{\cdot}10^{-2}{\cdot}\alpha - 1.04{\cdot}10^{-4}{\cdot}\gamma \\
& -8.65{\cdot}10^{-1}{\cdot}t - 4.67{\cdot}10^{-1}{\cdot}m; \ (HCLPF \geq 0 \text{ for all cases})
\end{aligned}
\tag{4}
$$

**Table 1.** Ranges of input and output of data.

| Input Features | Minimum | Maximum | Distribution |
|---|---|---|---|
| $c$ (kPa) | 30 | 120 | Uniform |
| $\varphi$ (deg.) | 10 | 60 | Uniform |
| $\alpha$ (deg.) | 10 | 70 | Uniform |
| $\gamma$ (kN/m$^3$) | 11 | 30 | Uniform |
| $t$ (m) | 1 | 5 | Uniform |
| $m$ | 0 | 1 | Uniform |
| **Output Features** | **Minimum** | **Maximum** | - |
| HCLPF (g) | 0 | 13.9 | - |

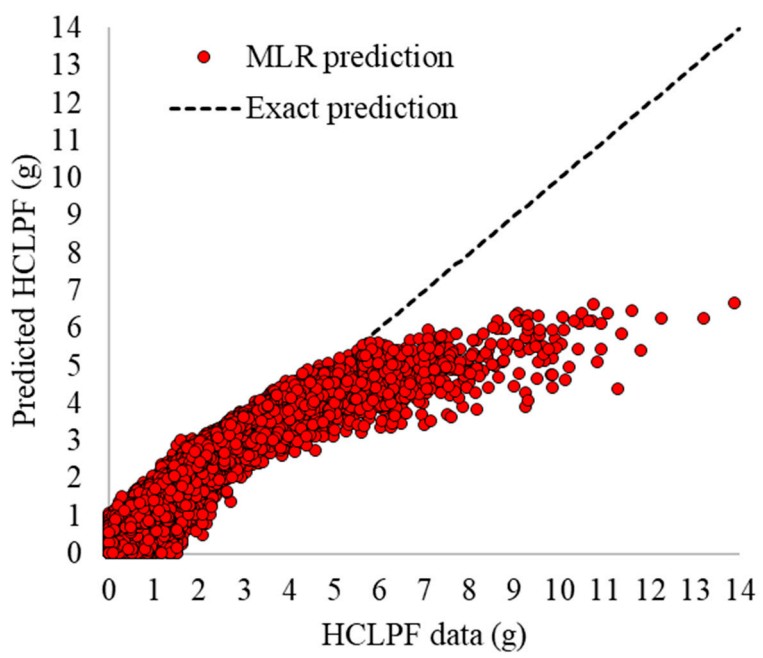

**Figure 5.** HCLPF seismic performance data of slopes with various conditions, and linear regression analysis results for the data [25,26].

The short explanation of the model obtained from the previous study is as below: As observed in Equation (4), the soil cohesion and internal friction angle showed a positive correlation with the HCLPF seismic performance, while a negative correlation was found regarding the slope angle, soil unit weight, normal thickness of slope failure planar and saturation ratio. These results of this equation show that the same tendency is exhibited not only on the slope of a specific condition of Figure 4 but also on various condition slopes within the considered range.

As shown in Figure 5, the obtained linear regression model [25,26] shows a significant correlation with the data, as this has an $R^2$ value of 0.8335. However, it can be seen that the deviation in the prediction accuracy of the model is considerably large, and the prediction accuracy is much lower in the high-value range of the HCLPF seismic performance. The reason for this seems attributed to the fact that the traditional linear regression analysis method cannot capture the nonlinear relationship between the slope of various conditions and the seismic performance, especially in the higher HCLPF seismic performance range. Therefore, in this study, we aim to develop a model that predicts such original data much better through machine learning methods. In the next section, we briefly introduce the concepts of machine learning methods that we used in this study.

### 2.3. Machine Learning Methods

2.3.1. Artificial Neural Network

An artificial neural network (ANN) is a computational model that simulates the knowledge acquisition and processing of human brain neurons. As the understanding of human neurons deepened in the early 20th century, a neuron model simulating these features was proposed [39]. As the fact that the change of the state of connection among neurons could occur was revealed (Hebb's Learning Rule) [40], that finding affected the previous neuron computational model, and a perceptron model with additional weightings was devised [41]. At present, based on the neuron and perceptron models, a multilayer perceptron (MLP) with greater expressive degrees of freedom through multiple layers of perceptron connections has been derived and researchers have frequently attempted to apply the approach to complex model learning in various fields [42].

The MLP structure can be divided into three parts: an input layer, a hidden layer, and an output layer. In the input layer, the number of neurons is determined by the number of input variables, and each neuron in this layer receives information from the outside and transmits the information to the hidden layer. The hidden layer defines the relationship between neurons in the input layer and expresses the complex association between input variables of the input layer. The output layer ultimately generates the results of the neural network for input variables based on information from the hidden layer. Here, all neurons between neighboring layers are completely connected to each other by weighting and bias. Specifically, the respective neurons of the hidden layer and the output layer multiply the input values by the weights, add the bias values and sum them, and then derive the sum at each layer through a transfer function. In this process, the MLP is called a fully connected feed-forward Neural Network, because the information is updated only in one direction from the input layer to the output layer.

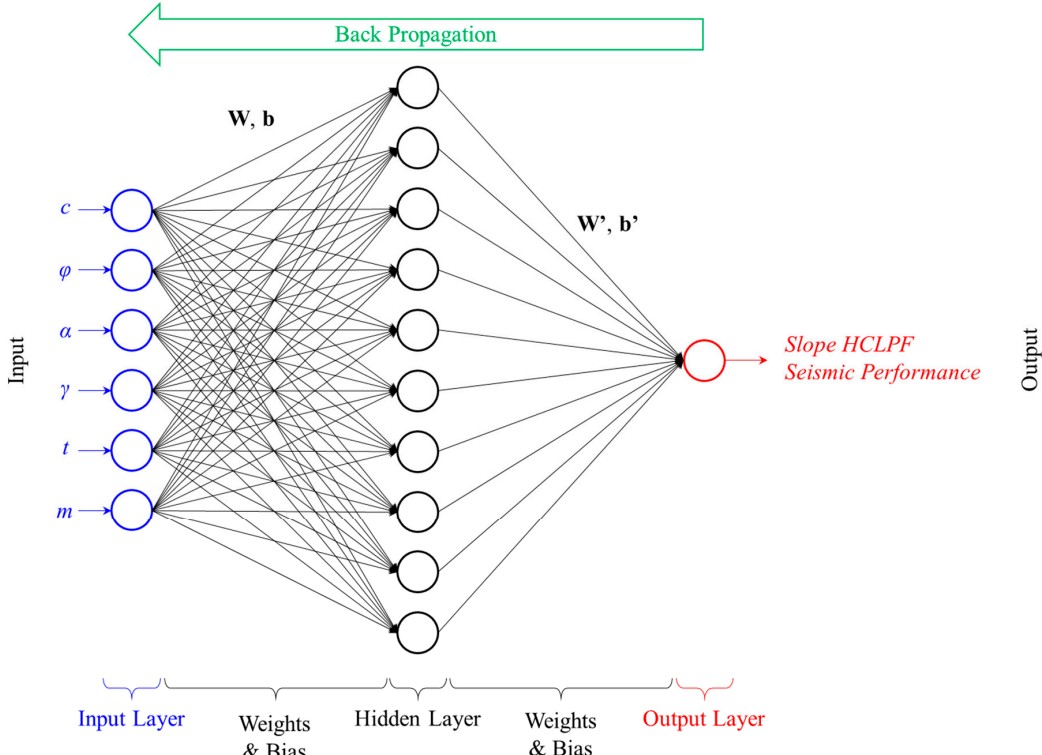

**Figure 6.** Artificial neural network model for slope seismic performance evaluation: W and W' represent the weighting factors for the input variables and b and b' indicate related coefficients.

The generated artificial neural network learns by changing the weight and deflection values of each neuron to minimize the difference between the output value predicted from the model and the actual output value and by utilizing the Gradient Descent Method. At this time, it is very difficult to change many weights of the neural network at the same time, so normally, the backpropagation (BP) technique is used to update only one layer at a time in the direction from the output layer to the input layer. The BP algorithm is a concept that makes the MLP learning practically possible, and research on a faster and more efficient optimization algorithm based on this is underway.

Figure 6 shows the ANN model developed in this study. This is the MLP of a two-layer feed-forward Neural Network, which consists of six main parameters that constitute the slope model mentioned in the previous section. A total of ten neurons were used for the hidden layer and one neuron for the output layer, and the final value of the output layer neuron means the seismic performance HCLPF of the slope. Here, the transfer function utilized the Sigmoid function at the hidden layer and the linear output function at the output layer. The number of neurons in the hidden layer was determined through a parametric study, because there is no general method to determine the optimal value. The Levenberg–Marquardt technique was used for the BP algorithm for ANN learning and the mean squared error was utilized as the error function expressing the ANN performance.

### 2.3.2. Support Vector Machine

The Support Vector Machine (SVM) is one of the machine learning methods widely used for data classification or regression analysis. It first started with Vladimir Vapnik and his colleagues [43]. The SVM regression analysis is a nonparametric technique that relies on kernel functions. In this study, we used the $\varepsilon$-SVM regression method utilizing L1 loss. In the $\varepsilon$-SVM regression method, the training data set consists of the input variable ($\mathbf{x}$) and the observed response ($y$), and the objective is to find a function $f(\mathbf{x})$ that simulates the data set well. Here, $f(\mathbf{x})$ is determined as a function that makes the difference between $y$ and $f(\mathbf{x})$ smaller than $\varepsilon$ and, at the same time, as flat as possible.

To be more specific, in the SVM regression analysis, the input variables are mapped into the n-dimensional feature space using the nonlinear mapping technique. The reason for this mapping is to transform the input data into a higher dimensional space to represent data with a high degree of the nonlinearity in a basic linear function form. This is expressed in mathematical form as follows:

$$f(\mathbf{x}) = \sum_{i=1}^{n} w_i \cdot \varphi_i(x) + b \tag{5}$$

Here, $\varphi_i(x)$ is a basic function set for the nonlinear mapping, $w_i$ is a weight value, and $b$ is a bias value. The fitness of the function for the input data is quantified as the following Loss Function, $L_\varepsilon$. In this study, $\mathbf{x} = [c, \varphi, \alpha, \gamma, t, m]$ and $y = [\text{HCLPF}]$.

$$L_\varepsilon(\mathbf{x}) = \begin{cases} 0 & \text{if } |y - f(\mathbf{x})| \le \varepsilon \\ |y - f(\mathbf{x})| & \text{otherwise} \end{cases} \tag{6}$$

In the SVM, the problem of finding the optimal model for the data based on the loss function defined above can be formulated as the following optimization problem that minimizes the loss function value while reducing the complexity of the model.

$$\text{minimize } \frac{1}{2}\|w\|^2 + C \sum_{i=1}^{n} \left( \xi_i + \xi_i^* \right)$$

$$\text{subject to : } \begin{cases} \forall i: \; y_i - f(\mathbf{x}) \le \varepsilon + \xi_i \\ \forall i: \; f(\mathbf{x}) - y_i \le \varepsilon + \xi_i^* \\ \forall i: \; \xi_i^* \ge 0 \\ \forall i: \; \xi_i \ge 0 \end{cases} \tag{7}$$

where $C$ is a positive numerical value and is a value that adjusts the penalty imposed on data points outside the range of $\varepsilon$ value, which helps prevent overfitting (regularization). $\xi_i$ and $\xi_i^*$ are introduced at each data point as slack variables introduced to deal with infeasible constraints.

The solution of the optimization problem in Equation (7) can be handled more simply through the Lagrange dual formulation as follows. The solution of the dual problem provides the lower bound of the solution of the primal problem of Equation (7). The optimal solutions of the primal and dual problems need not be the same, and the difference between these values is called the duality gap. However, if the optimization problem is convex and the constraint is satisfied, the optimal solution value of the primal problem can be obtained through the solution of the dual problem.

$$\text{minimize } \tfrac{1}{2}\sum_{i=1}^{N}\sum_{j=1}^{N}\left(\alpha_i - \alpha_i^*\right)\left(\alpha_j - \alpha_j^*\right)G\left(x_i, x_j\right) + \varepsilon\sum_{i=1}^{N}\left(\alpha_i + \alpha_i^*\right) - \sum_{i=1}^{N}\left(\alpha_i - \alpha_i^*\right)$$

$$\text{subject to :} \begin{cases} \sum_{i=1}^{N}\left(\alpha_i - \alpha_i^*\right) = 0 \\ \forall i : \ 0 \le \alpha_i \le C \\ \forall i : \ 0 \le \alpha_i^* \le C \end{cases} \tag{8}$$

Here, $\alpha_i$ and $\alpha_i^*$ are Lagrange multipliers with positive values, and the difference between the two values is called a support vector. $G(\mathbf{x})$ is a Gram matrix and can be expressed as:

$$G\left(x_i, x_j\right) = \left\langle \varphi(x_i), \varphi\left(x_j\right) \right\rangle \tag{9}$$

The Gram matrix $G(\mathbf{x})$ can be directly expressed as a kernel function such as a dot product, Gaussian, polynomial, etc., without having to know the transfer function in advance. Finally, the function model that predicts the new value can be expressed as follows:

$$f(x) = \sum_{i=1}^{N}\left(\alpha_i - \alpha_i^*\right)G(x_i, x) + b \tag{10}$$

In Equation (10), the support vector of $(\alpha_i - \alpha_i^*)$, $x_i$, and $b$ are the values that need to be obtained through the optimization method. The optimization can be obtained through general quadratic programming techniques, but decomposition methods, sequential minimal optimization (SMO), and iterative single data algorithm can be used for an efficient solution.

In this study, the following Gaussian functions were used as kernel functions, and the $C$ and $\varepsilon$ values were set to iqr($y$)/1.349 and iqr($y$)/13.49, respectively, where iqr($y$) represents the interquartile range values of the observed response $y$. The SMO method was used as the solver for optimization, and the final values for model parameters were also derived through this.

$$G\left(x_j, x_k\right) = \exp\left(-\left\|x_j - x_k\right\|^2\right) \tag{11}$$

### 2.3.3. Gaussian Process Regression

Gaussian process regression is one of the nonparametric techniques that utilize a kernel-based probabilistic model [44]. Generally, the GPR model for the input ($\mathbf{x}$) and the response ($y$) is expressed as follows:

$$y = \mathbf{h}(\mathbf{x})^T\boldsymbol{\beta} + \mathbf{f}(\mathbf{x}); \ \mathbf{f}(\mathbf{x}) \sim N(0, \mathbf{K}(\mathbf{x}, \mathbf{x}\prime; \theta)) \tag{12}$$

Here, $\mathbf{h}(\mathbf{x})$ is a vector consisting of basis functions, and it maps the original input variable to a multidimensional space. The $\boldsymbol{\beta}$ is a coefficient vector that is weighted to the basis function vector. The $\mathbf{f}(\mathbf{x})$ means a Gaussian process with a mean of "0" and a covariance of $k(x, x')$. $k(x, x')$ is a kernel function that can be expressed in various formats. The $\theta$ is the main parameter defining the kernel function. Specifically, the covariance matrix $\mathbf{K}(\mathbf{x}, \mathbf{x}')$ can be expressed as follows. In this study, $\mathbf{x}$ is

the six major variables of the slope model and $y$ is the corresponding HCLPF value representing the seismic performance of the slope.

$$\mathbf{K}(\mathbf{x}, \mathbf{x}\prime; \boldsymbol{\theta}) = \begin{pmatrix} k(x_1, x_1) & k(x_1, x_2) & \cdots & k(x_1, x_n) \\ k(x_2, x_1) & k(x_2, x_2) & \cdots & k(x_2, x_n) \\ \vdots & \vdots & \ddots & \vdots \\ k(x_n, x_1) & k(x_n, x_2) & \cdots & k(x_n, x_n) \end{pmatrix} \tag{13}$$

Based on the GPR model defined above, the probabilistic response of $y$ can be expressed as:

$$P(y|\mathbf{x}, \mathbf{f}) \sim N\left(\mathbf{h}^T \boldsymbol{\beta} + \mathbf{f}, \sigma^2 \mathbf{I}\right) \tag{14}$$

where $\mathbf{I}$ is the identity matrix and $\sigma$ indicates the standard deviation of the probability density function of $y$. To predict the response regarding a new input variable based on the constructed GPR probability model, the following three pieces of information are needed: (a) the coefficient vector $\boldsymbol{\beta}$ for the basis function vector, (2) the $\boldsymbol{\theta}$ values of the parameter for the covariance function $k(x, x')$, (3) the noise variance $\sigma^2$ value in the probability density function of $y$. In general, the method of estimating these parameter values is to find the parameter values of $\boldsymbol{\beta}$, $\boldsymbol{\theta}$, and $\sigma^2$ that maximize the likelihood of Equation (14), and this method is called Maximum Likelihood Estimation. This can be expressed as the following equations:

$$\boldsymbol{\beta}, \boldsymbol{\theta}, \sigma^2 = \underset{\boldsymbol{\beta}, \boldsymbol{\theta}, \sigma^2}{\text{argmax}} \log P\left(y|\mathbf{x}, \boldsymbol{\beta}, \boldsymbol{\theta}, \sigma^2\right) \tag{15}$$

$$\log P\left(y|\mathbf{x}, \boldsymbol{\beta}, \boldsymbol{\theta}, \sigma^2\right) = \begin{aligned} &-\tfrac{1}{2}\left(y - \mathbf{h}^T \boldsymbol{\beta}\right)^T \left[\mathbf{K}(\mathbf{x}, \mathbf{x}\prime; \boldsymbol{\theta}) + \sigma^2 \mathbf{I}\right]^{-1}\left(y - \mathbf{h}^T \boldsymbol{\beta}\right) \\ &-\tfrac{n}{2}\log 2\pi - \tfrac{1}{2}\log\left|\mathbf{K}(\mathbf{x}, \mathbf{x}\prime; \boldsymbol{\theta}) + \sigma^2 \mathbf{I}\right| \end{aligned} \tag{16}$$

Based on the parameter values estimated from the above method of maximum likelihood, the response ($y_*$) to the new input variable ($x_*$) can be mathematically predicted as follows:

$$P(y_*|y, \mathbf{x}, x_*) = N\left(y_* \middle| \mathbf{h}(x_*)^T \boldsymbol{\beta} + \boldsymbol{\mu}, \sigma_*^2 + \sum\right) \tag{17}$$

Here, the mean ($\boldsymbol{\mu}$) and covariance matrices ($\boldsymbol{\Sigma}$), which define the probability model of Equation (17), are described as follows.

$$\boldsymbol{\mu} = \mathbf{K}(x_*, \mathbf{x})\left(\mathbf{K}(\mathbf{x}, \mathbf{x}\prime) + \sigma^2 \mathbf{I}\right)^{-1}\left(y - \mathbf{h}^T \boldsymbol{\beta}\right) \tag{18}$$

$$\sum = k(x_*, x_*) - \mathbf{K}(x_*, \mathbf{x})\left(\mathbf{K}(\mathbf{x}, \mathbf{x}\prime) + \sigma^2 \mathbf{I}\right)^{-1}\mathbf{K}(\mathbf{x}, x_*) \tag{19}$$

In this study, we used the constant value "1" as the basis function and the kernel function used the squared exponential kernel with a separate scale per predictor.

$$k\left(x_i, x_j|\boldsymbol{\theta}\right) = \sigma_f^2 \exp\left[-\frac{1}{2}\sum_{m=1}^{d} \frac{\left(x_{im} - x_{jm}\right)^2}{\sigma_m^2}\right] \tag{20}$$

Here, the parameter $\theta$ that determines the characteristics of the kernel function is composed of $\sigma_f$ and $\sigma_m$, and $d$ is the total number of input variables $x$. As a fitting method to find the parameters, we used the subset of data points approximation technique because of the large amount of data ($n > 2000$).

## 3. Results and Discussion

In order to evaluate the accuracy of the predictive models obtained from the machine learning method, the following four performance indicators were used in this study [45]. The linear correlation

coefficient ($R$) is a measure of the linear relationship between the predicted value of the proposed model and the actual data and has a value of "+1" to "−1."

$$R = \frac{\sum_{i=1}^{n} (y_i - \overline{y})(y\prime_i - \overline{y\prime})}{\sqrt{\sum_{i=1}^{n}(y_i - \overline{y})^2} \sqrt{\sum_{i=1}^{n}(y\prime_i - \overline{y\prime})^2}} \tag{21}$$

Here, $y$ is an actual value, and $y\prime$ represents a predicted value. $n$ denotes the total number of data samples, and $\overline{y}$ and $\overline{y\prime}$ show the average of the actual and predicted values. If $R$ has a value of +1, then both datasets represent a perfect amount of fit, which means that the actual and predicted values have the same value. The coefficient of determination ($R^2$) has a value between "0" and "1," and represents the degree of the strength of a linear relationship between the actual value and the model-predicted value. Statistically, this indicates the portion that can be explained by the applied probability model among the variation of input variables. $R^2$ is mathematically expressed as follows.

$$R^2 = 1 - \frac{\sum_{i=1}^{n}(y_i - y\prime_i)^2}{\sum_{i=1}^{n}(y_i - \overline{y})^2} \tag{22}$$

The higher the $R^2$ value, the larger the predictive power of the proposed model. The $R$ and $R^2$ have a strong correlation with each other. When the squared error loss between the actual value and the model-predicted value is small, the $R$-value has the same value as the square root of $R^2$. The mean absolute error ($MAE$) and the root-mean-square error ($RMSE$) are indicators for quantitatively showing how the predicted values differ from the actual values. This is based on the absolute value of the difference or the root of the square of the difference between actual values and model-predicted values. Mathematically, we can express these as follows:

$$MAE = \frac{\sum_{i=1}^{n}|y_i - y\prime_i|}{n} \tag{23}$$

$$RMSE = \sqrt{\frac{\sum_{i=1}^{n}(y_i - y\prime_i)^2}{n}} \tag{24}$$

The characteristics of these two indicators mean that the smaller the value, the better the prediction ability of the model.

To derive the HCLPF seismic performance prediction model of the slope, 10,000 items of data (see Section 2) were randomly divided into training and testing samples. To be more specific, 90% of the data, i.e., 9000 samples, were utilized to derive the predictive model by assigning these to the training set, and the remaining 10% of the data, i.e., 1000 samples, were used to verify the accuracy (effectiveness) of the derived predictive model. The reason for the 90–10 division of these data is to cross-validate the derivation model within the data. This cross-validation is a method that can be applied to models and data sets to estimate out-of-sample errors. Specifically, it is well known that estimating the resulting prediction error by using the same data which are used for deriving the model will produce overly optimistic results. As such, it is common to test the model on a new data set to better estimate the out-of-sample prediction error. This can be realized by a $k$-fold cross-validation, and one of the most commonly used methods is 10-fold cross-validation [46]. Therefore, in this study, data were basically classified according to the 10-fold cross-validation method.

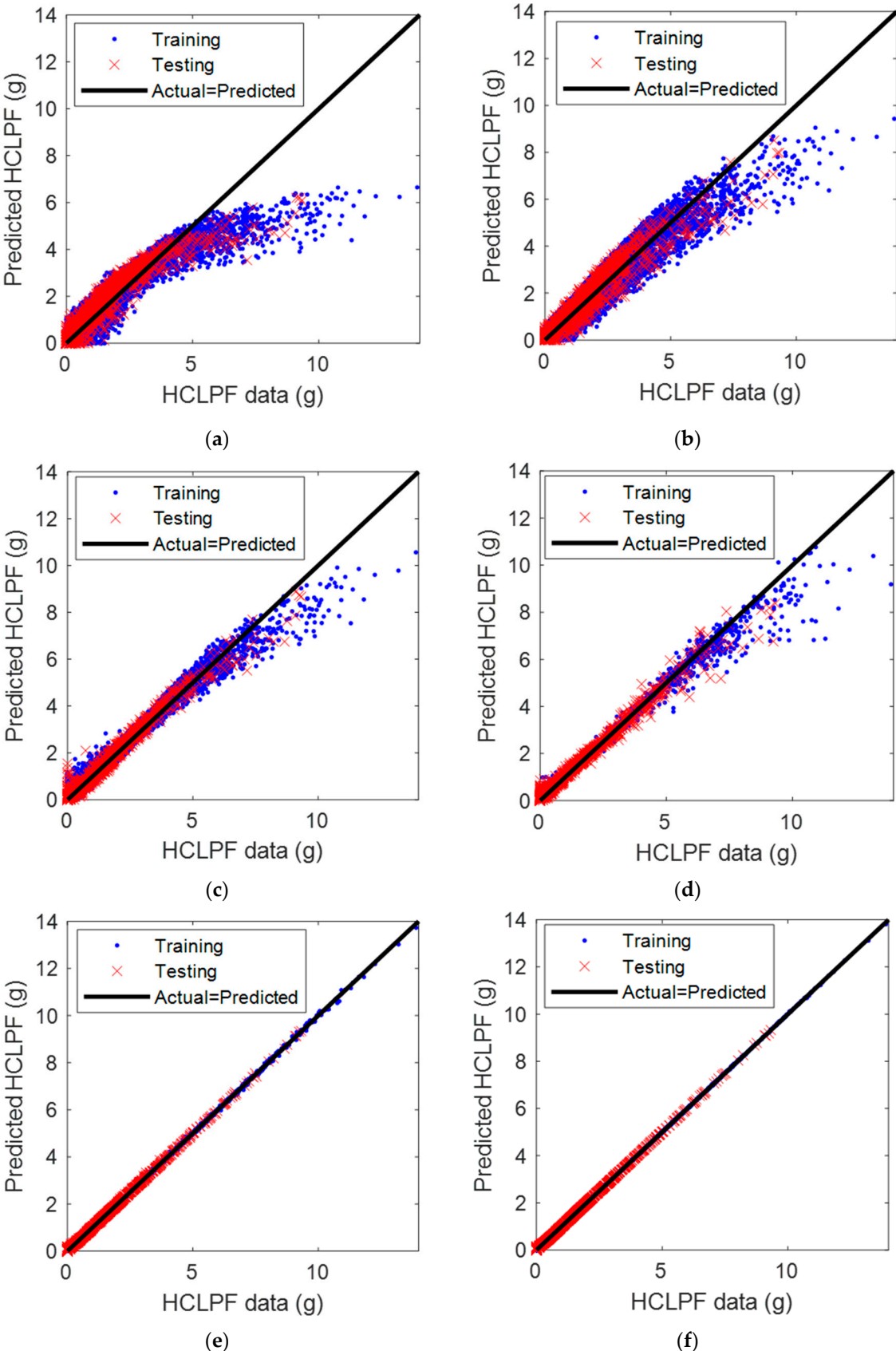

**Figure 7.** Actual versus predicted HCLPF seismic performance of slopes: (**a**) MLR; (**b**) MLR w/int.; (**c**) MPR; (**d**) SVM; (**e**) ANN; (**f**) GPR.

To verify the performance validity of the predictive model through the machine learning method introduced in this study, the general statistical regression methods Multiple Linear Regression (MLR) and Multiple Polynomial Regression (MPR) were applied to the identical data sets, and compared with those obtained through machine learning methods. For the MLR model, a general model that does not take into account the interaction among input variables and a model that considers this (MLR with interaction: MLR w/int.) were applied separately. Here, the general MLR model is the same model utilized for the linear regression analysis of the previous study (see Section 2.2). For the MPR model, a nonlinear model was constructed based on the second-order polynomial and the parameters related to the model were estimated through optimization techniques.

Based on the six major input variables of the slope mentioned above and the HCLPF seismic performance data, the predicted values of the model derived from the general statistical regression methods and the machine learning methods and their corresponding actual values are shown together in Figure 7. Table 2 compares the performance indices among the models. The first row of Table 2 represents the dataset used, performance indicators, and models derived from each method. The training set of the dataset was used to derive the model, and the testing set was used to verify the derived model. Table 2 shows in detail the results of calculating the performance index of models derived from each dataset. This shows that the performance indicators of the models derived from the training set and the testing set are not significantly different. This seems that the derived models show satisfactory performance in the new dataset that is not used for the training because these are not overfitted to the training dataset. Also, it can be confirmed that the models using machine learning methods show superior performance compared to the traditional statistical regression method. Figure 8 shows a radial graph that clearly compares the results of each performance index shown in Table 2. The detailed findings and specific discussions obtained from these Figures and Table 2 are described as below:

- The MLR-based model tends to predict the actual value of the input variables well on the output values between HCLPF 0–3 g, but the variance of the predictive values is large. Further, for HCLPF values 3 g or more, a value smaller than the actual value is predicted, and our results confirmed that the prediction ability was remarkably deteriorated.
- The MLR-based model considering the interaction among input variables simulates the actual values of input variables well on average regarding the output values between the HCLPF values of 0–6 g, but the variance of these results is still large. Additionally, we confirmed that the model yields a biased value that is less than the actual value for an HCLPF value of 6 g or more.
- The MPR model predicts the actual values for the input variables well for output values of HCLPF between 0–6 g, and the variance of the predicted values is relatively small compared with the existing MLR model. However, in the value range of more than HCLPF 6 g, a biased value smaller than the actual value is still predicted.
- The SVM model predicts the actual values of the input variables well for the output values between the HCLPF values of 0–6 g compared with the MPR model, and the variance of these predicted values is also relatively small. However, it still does not show a satisfactory accuracy in the range of more than an HCLPF of 6 g.
- The ANN and GPR models predict the actual value of the input variables almost exactly in the range of all the HCLPF values compared with the previously mentioned models, and the variance of such predicted values is also quite small. The ANN and GPR models show similar predictive performances, but the GPR model predicts more closely to the actual values than the ANN does. That result is quantitatively observed in Table 2 and Figure 8.

The observations and analogies confirmed in each model above can be more clearly seen when we compare the distributions on the differences between the HCLPF actual values and the predicted values of each model as shown in Figure 9. When we compare the GPR model with other models, the distribution of the difference between the actual and predicted HCLPF values concerning the GPR

model is concentrated on a value of '0'. This indicates that the GPR model predicts a value closer to the actual value probabilistically than the other model, which means that it is the model that predicts the HCLPF seismic performance of the slope most accurately.

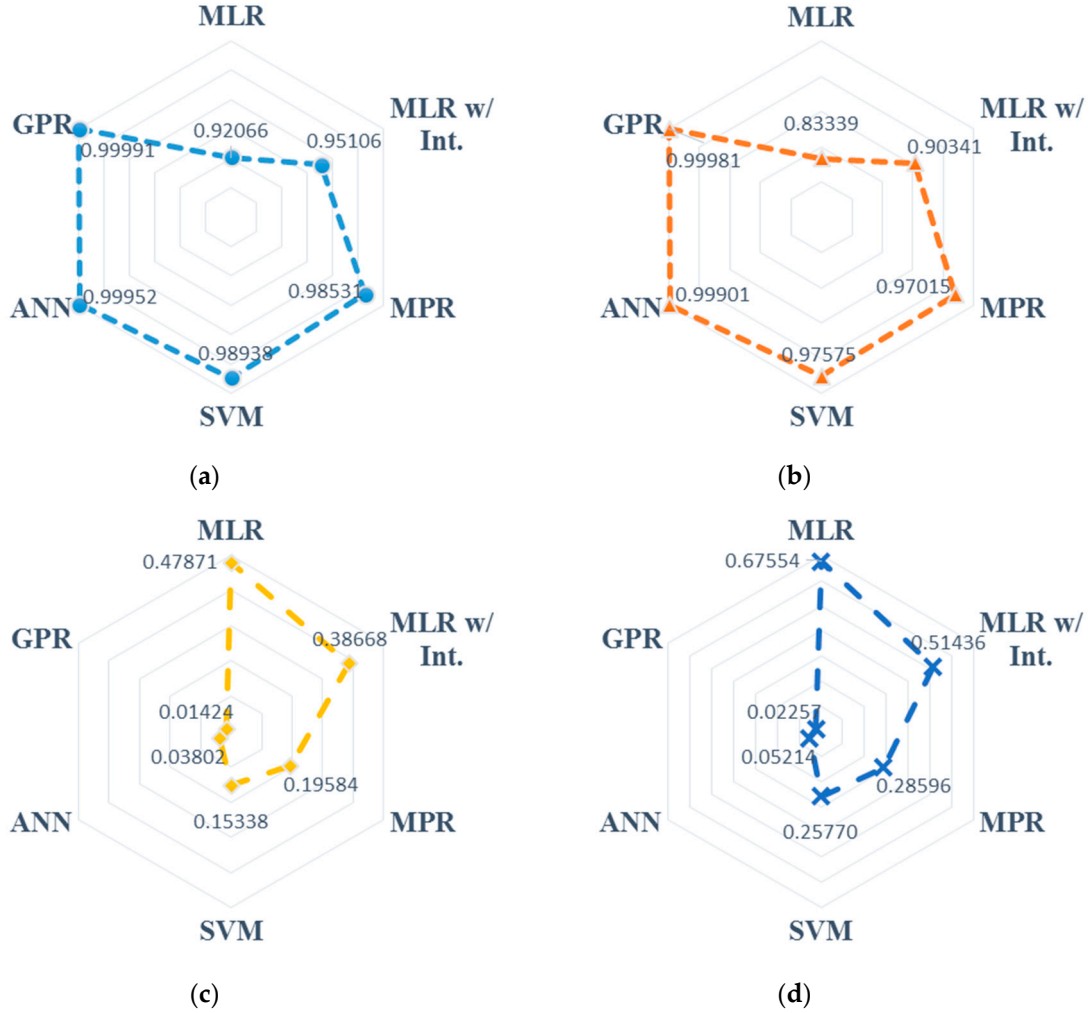

**Figure 8.** Performance measure results of the predicted models based on testing samples: (**a**) *R*; (**b**) $R^2$; (**c**) *MAE*; (**d**) *RMSE*.

**Table 2.** Performance comparison of the predicted models using statistical and machine learning methods.

| Dataset | Performance Measures | MLR | MLR w/Int. | MPR | SVM | ANN | GPR |
|---|---|---|---|---|---|---|---|
| Training Set | *R* | 0.92120 | 0.95403 | 0.98552 | 0.99376 | 0.99958 | 0.99994 |
| | $R^2$ | 0.83428 | 0.90927 | 0.97092 | 0.98539 | 0.99914 | 0.99988 |
| | *MAE* | 0.46368 | 0.37508 | 0.19443 | 0.11448 | 0.03682 | 0.01266 |
| | *RMSE* | 0.71673 | 0.53032 | 0.30023 | 0.21281 | 0.05153 | 0.01927 |
| Testing Set | *R* | 0.92066 | 0.95106 | 0.98531 | 0.98938 | 0.99952 | 0.99991 |
| | $R^2$ | 0.83339 | 0.90341 | 0.97015 | 0.97575 | 0.99901 | 0.99981 |
| | *MAE* | 0.47871 | 0.38668 | 0.19584 | 0.15338 | 0.03802 | 0.01424 |
| | *RMSE* | 0.67554 | 0.51436 | 0.28596 | 0.25770 | 0.05214 | 0.02257 |

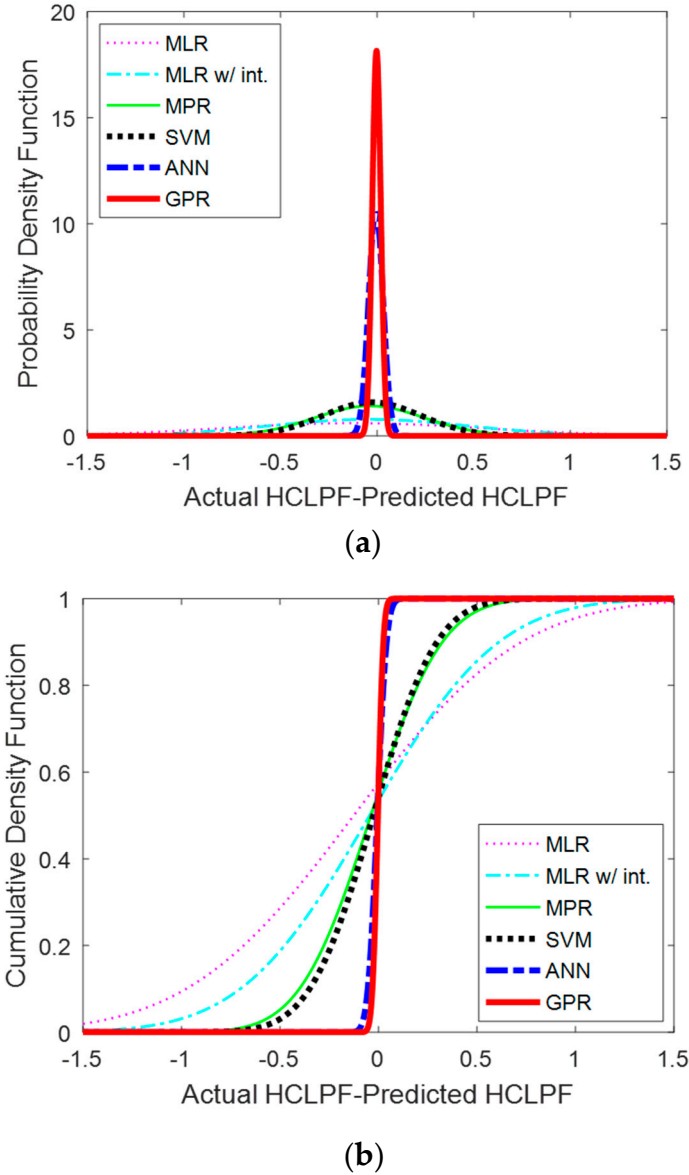

**Figure 9.** Distribution of actual HCLPF value minus predicted HCLPF value for six different models on testing samples: (**a**) probabilistic density function; (**b**) cumulative density function.

The reason for this result is that the MLR and MPR derive a fit model by finding the parameters of the structured function which simulates the actual data in the low dimensional linear input variable space. On the other hand, the machine learning approaches, such as SVM, ANN, and GPR, basically change the dimension of the input variable into a high-dimensional input space which can well simulate the data. Then, the methods find the parameter values of the basic model function used. Due to the characteristics of these machine learning methods, the models SVM, ANN, and GPR represent the actual data more flexibly than the previous MLR and MPR models. Also, the large performance difference between the traditional statistical models and machine learning models is attributed to the fact that the HCLPF slope seismic performance data have a high degree of nonlinearity. These results are similar to the results of a study looking for a machine learning model that estimates the frictional capacity of a driven pile in the clay soil [29]. In that study, the GPR model showed superior performance compared to other models such as ANN, etc. Also, in the recent study attempting to develop an efficient model for predicting high-performance concrete compressive strength, it was seen that the GPR model produced better prediction results than the models obtained by other methods [47].

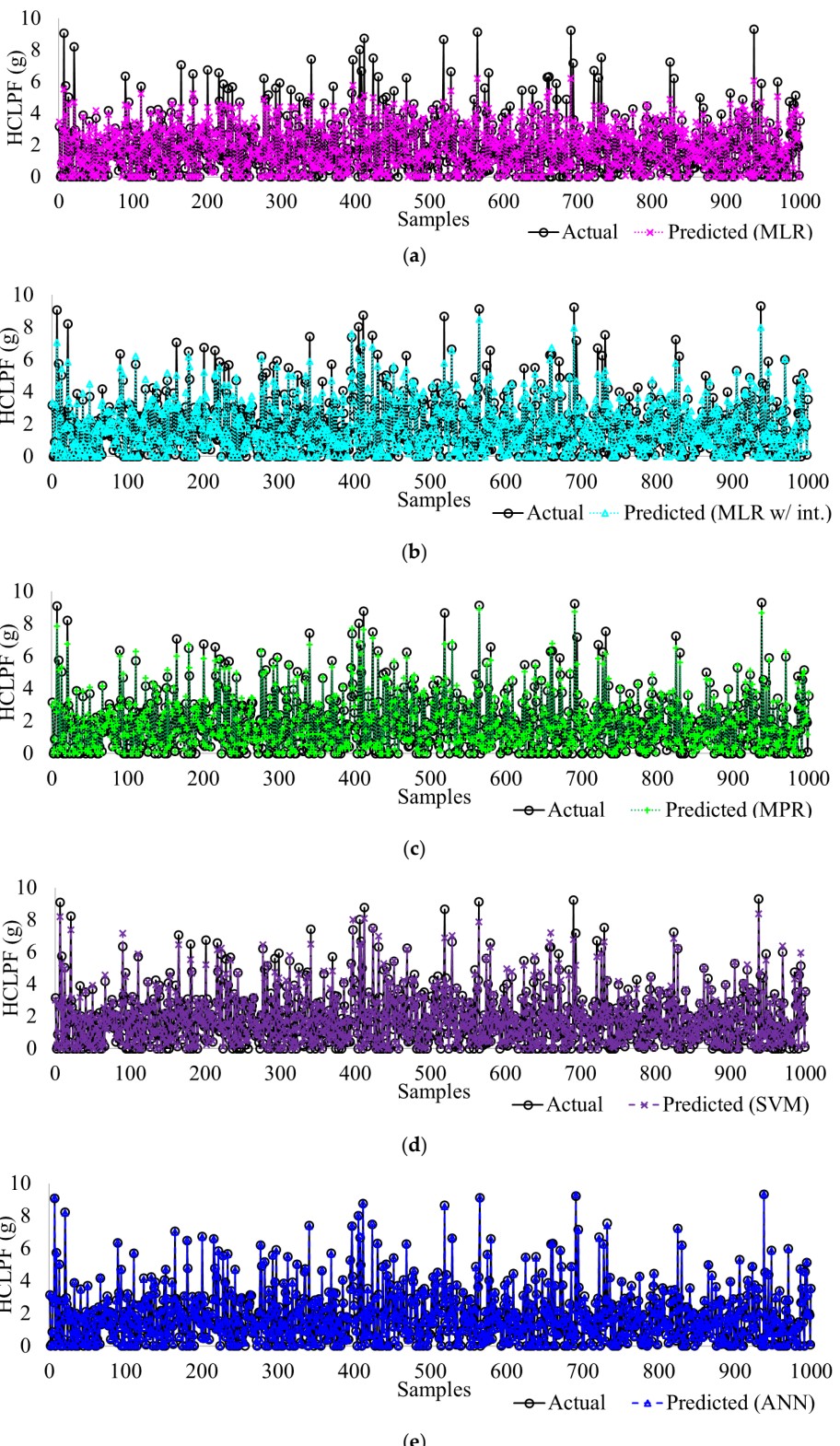

**Figure 10.** *Cont.*

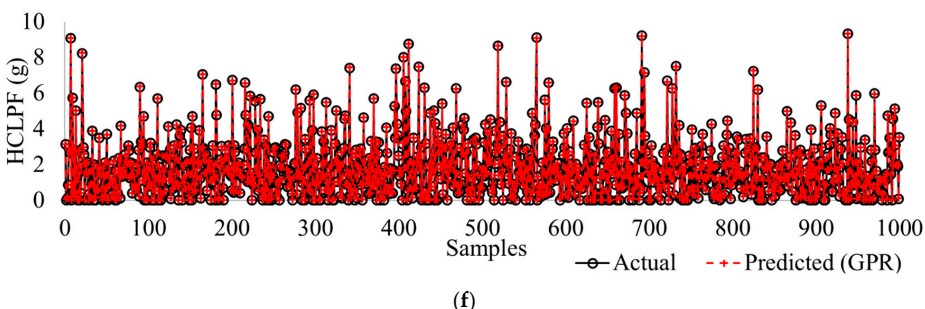

(**f**)

**Figure 10.** Actual and predicted HCLPF seismic performance of slopes with respect to testing samples; (**a**) MLR; (**b**) MLR w/int.; (**c**) MPR; (**d**) SVM; (**e**) ANN; (**f**) GPR.

Finally, Figure 10 shows the predicted value and actual value of each model concerning testing data. Here, the x-axis represents the sample index in a total of 1000 test data samples, and the y-axis represents the HCLPF seismic performance of the slope. As the figure confirms, the GPR and ANN models can describe actual values of data better than the MLR, MLR w/int., and MPR models. For the SVM model, it gives a more accurate value than the MLR and MLR w/int. models, but is a little bit better or there is no difference compared to the performance of the MPR model.

## 4. Conclusions

This study developed models that predict probabilistic seismic performances of slopes relatively accurately and efficiently through various machine learning methods. The validity of the obtained models was verified by comparing results for the proposed models with the results of the conventional multilinear and multinonlinear regression models. As machine learning methods for predicting slope seismic performance, three methods, SVM, ANN, and GPR, which are widely used in geotechnical engineering, were used. To be more specific, the data used in this study were obtained from the seismic fragility analysis for the slope and was a 5% failure probability of the 95% confidence interval (i.e., HCLPF) of the slope seismic fragility curve as the representative performance value. For the slope seismic fragility analysis, the empirical model based on the Newmark permanent displacement method was used for the analysis of the stability of the slope caused by the earthquake. The limit state for the permanent displacement defining the slope failure was estimated through correlation data between the observed slope failures in a previous earthquake and permanent displacement predicted using related slope properties.

As a result, we drew up the following conclusions through a comparison study between the prediction model of the statistical MLR method of the previous study and the prediction models of the machine learning methods. The MLR model ($R^2 = 0.83$, $RMSE = 0.68$) of the previous study, on average, predicted values well in relatively low range values of the HCLPF seismic performance of the slope, but the variance of the predicted values was very large. Also, it did not provide satisfactory results in the high-value range of HCLPF seismic performance of the slope. The MLR w/int. model ($R^2 = 0.90$, $RMSE = 0.51$) and MPR model ($R^2 = 0.97$, $RMSE = 0.29$) were able to increase the accuracy of the predicted values by increasing the degree of the dimension of a basic function used in the MLR model. However, the variance of the predicted values was still high and inaccurate values were obtained in the high-value range of the HCLPF seismic performance of the slope. The SVM model ($R^2 = 0.98$, $RMSE = 0.26$) improved the accuracy of prediction and reduced the dispersion in the predicted values by adjusting the dimension of the input variables. However, in the case of the high-value range of the HCLPF seismic performance of the slope, deviations still existed and the prediction accuracy of the seismic performance could not be significantly improved compared with the MPR model. The ANN ($R^2 = 0.9990$, $RMSE = 0.0521$) and GPR ($R^2 = 0.9998$, $RMSE = 0.0226$) models showed the best predictive performance compared to the other previous models. This indicated that such models could well predict the HCLPF seismic performance values of the slope in all

the domains. If comparing the results of the ANN model and the GPR model, we confirmed that the GPR model predicted a value slightly closer to the actual value.

The prediction models of the HCLPF seismic performance of the slope obtained from machine learning methods in this study can be used as an analysis tool to efficiently and accurately evaluate the seismic safety of the slope in future earthquakes. We also expect that this model can be integrated into the development of a seismic vulnerability map for all slope spatially distributed. In the future research, it is judged to be necessary to develop a predictive model by applying more advanced machine learning methods to these slope seismic performance data, and to compare the obtained model with the results of the models obtained from this study (i.e., the models which only utilized the individual learning algorithms). The advanced machine learning methods can include, but not be limited to, the ensemble and the hybrid machine learning methods.

**Author Contributions:** Conceptualization, S.K., D.H., M.K., and S.E.; Methodology, S.K. and D.H.; Validation, S.K. and S.E.; Formal Analysis, S.K. and S.E.; Data Curation, S.K. and S.E.; Writing—Original Draft Preparation, S.K.; Writing—Review & Editing, S.K., D.H., M.K., and S.E.; Funding Acquisition, D.H. and M.K. All authors have read and agreed to the published version of the manuscript.

**Funding:** This work was funded by the National Research Foundation of Korea (NRF) grant funded by the Korea government (Ministry of Science and ICT) (No. 2020R1G1A1005510). Also, this work was funded by the National Research Foundation of Korea (NRF) grant funded by the Korea government (NRF-2017M2A8A4015290).

**Acknowledgments:** This work was supported by the National Research Foundation of Korea (NRF) grant funded by the Korea government (Ministry of Science and ICT) (No. 2020R1G1A1005510). Also, this work was supported by the National Research Foundation of Korea (NRF) grant funded by the Korea government (NRF-2017M2A8A4015290).

**Conflicts of Interest:** The authors declare no conflict of interest.

## Abbreviations

| Acronyms | Description |
|---|---|
| SVM | Support vector machine |
| ANN | Artificial neural network |
| GPR | Gaussian process regression |
| HCLPF | The high-confidence-low-probability-of-failure |
| MLP | Multi-layer perceptron |
| SMO | Sequential minimal optimization |
| CL | Confidence Level |
| $R$ | The linear correlation coefficient |
| $R^2$ | The coefficient of determination |
| $MAE$ | Mean absolute error |
| $RMSE$ | Root mean squared error |
| MLR | Multiple linear regression |
| MPR | Multiple polynomial regression |
| MLR w/int. | Multiple linear regression with interaction |

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
