# Peer review of "Development of a Probabilistic Seismic Performance Assessment Model of Slope Using Machine Learning Methods"

_sustainability, doi:10.3390/su12083269_

Round 1

Reviewer 1 Report

The paper describes an approach to the seismic performance assessment model of slope problem using machine learning techniques. It is well written and only minors were found (see below). The paper is fine regarding the state of the art, methods and results. Its novelty is not in the proposal of some new technique, it is in the proposal of using machine learning techniques to the problem at hand. The results are very good, in particular for ANN and GPR.

Just one question to the authors: the most common division between train and test set is 70/30. Why did the authors used 90/10? Did they tried other division ratios?

Minor erros:
L22 - non-linearistic -> non-linear
L27 - of the previous study -> of previous studies
L72 - developed method -> developed the method OR developed a method
L200 - the Figure 4 -> remove "the"
Figure 4 - R^2 is used in the figure but R2 is used in the main text
L215 - The MLR acronym should be introduced here because it is used in Figure 5
L262 - Decent -> Descent
L274 - there was no -> there is no
L371 - see section 2) -> Section

Author Response

The paper describes an approach to the seismic performance assessment model of slope problem using machine learning techniques. It is well written and only minors were found (see below). The paper is fine regarding the state of the art, methods and results. Its novelty is not in the proposal of some new technique, it is in the proposal of using machine learning techniques to the problem at hand. The results are very good, in particular for ANN and GPR.

Author’s Reply:

We sincerely thank the reviewer for taking the time to review our manuscript. Also, we really appreciate that the reviewer conceded a certain degree of merit concerning this study. We revised the original manuscript based on the reviewer comments described below. Such revision works clearly helped to improve the quality of the manuscript. We appreciated your comments, again.

Just one question to the authors: the most common division between train and test set is 70/30. Why did the authors used 90/10? Did they tried other division ratios?

Author’s Reply:

Thank you for the reviewers' in-depth comments. As the reviewer pointed out, in the original manuscript, there was no discussion of why this study divided the data into the 90-10 form to create a predictive model. Therefore, the paper was revised by adding these discussions. For the detailed description, please refer to the related contents of section 3.

Also, as the reviewer mentioned, we can change the 90-10 data division to the 70-30 data division (i.e., one of the commonly used methods), and we can also apply this division form to this study. However, the results of different ratios of data division also gave almost the same results as those of the 90-10 data division. Therefore, this paper was mainly described based on the results of the 90-10 data division.

Minor erros:

L22 - non-linearistic -> non-linear

Author’s Reply:

As suggested by the reviewer, we corrected the term. For the details, please refer to the related part.

L27 - of the previous study -> of previous studies

Author’s Reply:

As suggested by the reviewer, we corrected the phrase. For the details, please refer to the related part.

L72 - developed method -> developed the method OR developed a method

Author’s Reply:

As suggested by the reviewer, we corrected the phrase. For the details, please refer to the related part.

L200 - the Figure 4 -> remove "the"

Author’s Reply:

As suggested by the reviewer, we removed “the”. For the details, please refer to the related part.

Figure 4 - R^2 is used in the figure but R2 is used in the main text

Author’s Reply:

As suggested by the reviewer, we consistently used “R22” expression as an exponential form in all figures, tables, and main texts.  

L215 - The MLR acronym should be introduced here because it is used in Figure 5

Author’s Reply:

As pointed by the reviewer, we introduced the MLR acronym here in this line of text that the reviewer noted. Also, we made an acronyms table for all the abbreviations used in this study.    

L262 - Decent -> Descent

Author’s Reply:

As pointed by the reviewer, we corrected the typo.

L274 - there was no -> there is no

Author’s Reply:

As suggested by the reviewer, we corrected the phrase. For the details, please refer to the related part.

L371 - see section 2) -> Section

Author’s Reply:

As pointed by the reviewer, we corrected this.

Reviewer 2 Report

The objective of this study is to propose a model that can predict the seismic performance of slope relatively accurately and efficiently by using the machine learning technique.

In first paragraph authors should provide some relevant references to given informations.

Which "several resrechers"? Avoid using this and state references.

Introduction misses some of the key literatire overview. For example, lines 91-110?

What is scope and aim of this study stated in Introduction-it is missing? Could some part of Introduction be cutted and put into follwoing chapters. Introduction is too big, last paragraph unncessary. 

Sentences are repeating in Introduction and chapter 2. avoid this. 

Figure 1b, font is too small. Figure 2-reference? Figure 3-can you put grid lines?Figure 4-for R2 use two decimals.

Why use linear regression in Figure 5? Power law maybe? 2nd order regression? 

In text for R^2 shoule be in exponent.

Equations format should be unified.

Please check page 12, chapter 2, some symbols should be corrected. R^2. Generally, this part is not needed since it is known from statistics (equations) but authors choose if they want them. But please check formulas and symbols.

Figure 7a,b maybe linear regression is not good?

Figure 10, font is too small.

Could autohrs disscuss their results with some similar studies? This is missing.

Author Response

The objective of this study is to propose a model that can predict the seismic performance of slope relatively accurately and efficiently by using the machine learning technique.

Author’s Reply:

First of all, we thank the reviewer for taking the time to review our manuscript. We revised the original manuscript based on the reviewer comments described below. Such a revision work significantly helped to improve the quality of the manuscript. We sincerely appreciate your comments.

In first paragraph authors should provide some relevant references to given informations.

Author’s Reply:

Thank you for the reviewer's comments. We revised the manuscript by providing the relevant references regarding the given information. For the detailed information, please refer to the related contents of the first paragraph of Section 1.

Which "several resrechers"? Avoid using this and state references.

Author’s Reply:

As pointed by the reviewer, we avoided the phrase “several researchers” and cited the references associated with these in the text of the manuscript. For the details, please refer to the related contents.

Introduction misses some of the key literatire overview. For example, lines 91-110?

Author’s Reply:

We overall revised the section of the Introduction. We had this revision work included adding some of the key literature overview regarding the paragraph that the reviewer mentioned.

What is scope and aim of this study stated in Introduction-it is missing? Could some part of Introduction be cutted and put into follwoing chapters. Introduction is too big, last paragraph unncessary.

Author’s Reply:

We overall revised the section of the Introduction. We clearly described the scope and the purpose of this study. We removed the unnecessary paragraph according to the reviewers' suggestion. For the detailed description, please refer to the Introduction section of the revised manuscript.

Sentences are repeating in Introduction and chapter 2. avoid this.

Author’s Reply:

According to the reviewer's comment, we removed the repeating contents in Section 2.

Figure 1b, font is too small. Figure 2-reference? Figure 3-can you put grid lines?Figure 4-for R2 use two decimals.

Author’s Reply:

The followings were revised according to the reviewer suggestions:

  • We increased the font size in Figure 1b.
  • We added the reference to Figure 2.
  • We put the grid lines in Figure 3.
  • We used two decimals for R2 in Figure 4.

Why use linear regression in Figure 5? Power law maybe? 2nd order regression?

Author’s Reply:

The previous study basically used a multiple linear regression (MLR) analysis for model development. Thus, in Figure 5, we presented such a MLR model result together with the original data. Regarding the 2nd order regression, in this study, we applied this regression form to the original data in the course of the model development process. The power law was not used for this study.  

In text for R^2 shoule be in exponent.

Author’s Reply:

In the manuscript, we corrected R2 expression in the exponent form.  

Equations format should be unified.

Author’s Reply:

We checked all equation formats in the manuscript and corrected the wrong equation formats. For the detailed modification, please refer to the revised manuscript.  

Please check page 12, chapter 2, some symbols should be corrected. R^2. Generally, this part is not needed since it is known from statistics (equations) but authors choose if they want them. But please check formulas and symbols.

Author’s Reply:

We overall checked symbols, R2, etc. including those on page 12 of the manuscript and corrected the wrong formulas and symbols.      

Figure 7a,b maybe linear regression is not good?

Author’s Reply:

Relatively, the MLR model results overall did not provide a good performance compared to the other models obtained from MPR and machine learning methods. For the detailed discussion, please refer to the related contents in Section 3.      

Figure 10, font is too small.

Author’s Reply:

We revised Figure 10 for improving the quality of the Figure. Such a revision included increasing the font size of Figure 10.      

Could autohrs disscuss their results with some similar studies? This is missing.

Author’s Reply:

We appreciated the in-depth comment of the reviewer. We discussed the results of this study based on similar studies recently conducted. For the detailed description, please refer to the related contents of Section 3.

Reviewer 3 Report

This paper computes the response of slopes, in terms of HCLPF, under varied input parameters. The computed HCLPF values are fitted using various machine learning methods and the accuracy of the different models are compared. Gaussian Process Regression and Artificial Neural Networks performed better in terms of capturing the median HCLPF with lesser standard deviation. Since performing a slope displacement analysis using the Newmark method for thousands of random input parameters is time consuming, developing machine learning models are a good alternative. Therefore, there may be value in this work.

Please consider the following comments.

  • How did you generate the slope displacement data given input parameters? Did you use equations (1)-(3)? You have to be very clear about this. If you used the empirical equation, how is its accuracy compared to a full Newmark displacement analysis? Explain these more clearly in the paper.
  • There is some redundant text in the paper that can be removed. For example, the explanation about R2 in lines 229-236 can be excluded. The discussion about R, R2, etc. between lines 350-367 can be shortened. 
  • Overall, the language is good. Some places may require english improvements. Please check again.

Author Response

This paper computes the response of slopes, in terms of HCLPF, under varied input parameters. The computed HCLPF values are fitted using various machine learning methods and the accuracy of the different models are compared. Gaussian Process Regression and Artificial Neural Networks performed better in terms of capturing the median HCLPF with lesser standard deviation. Since performing a slope displacement analysis using the Newmark method for thousands of random input parameters is time consuming, developing machine learning models are a good alternative. Therefore, there may be value in this work. Please consider the following comments.

Author’s Reply:

We sincerely thank the reviewer for taking the time to review our manuscript. Also, we really appreciate that the reviewer acknowledged merit concerning this study. We revised the original manuscript based on the reviewer comments described below. Such revision works helped to improve the quality of the manuscript. We appreciated your comments, again.

  • How did you generate the slope displacement data given input parameters? Did you use equations (1)-(3)? You have to be very clear about this. If you used the empirical equation, how is its accuracy compared to a full Newmark displacement analysis? Explain these more clearly in the paper.

Author’s Reply:

We appreciated the reviewers' in-depth comments. As the reviewer pointed out, in the original manuscript, there were no explanations of how we generate the slope displacement data concerning input parameters, and how the empirical equation is accurate. Thus, the paper was revised by adding these discussions. For the detailed description, please refer to the related contents of subsection 2.2 (for the slope displacement data generation) and subsection 2.1 (for the model accuracy).

  • There is some redundant text in the paper that can be removed. For example, the explanation about R2 in lines 229-236 can be excluded. The discussion about R, R2, etc. between lines 350-367 can be shortened.

Author’s Reply:

As suggested by the reviewer, we removed the redundant text for the R2. Thanks for your comment.

  • Overall, the language is good. Some places may require english improvements. Please check again.

Author’s Reply:

As suggested by the reviewer, we checked all sentences for English improvements. Thus, the original manuscript was revised based on such a check. For the details, please refer to the overall revised manuscript.

Reviewer 4 Report

This paper presents a comparative analysis of statistical and machine learning methods for probabilistic seismic performance assessment. The paper has merit however many revisions are essential. 

  1. 1. The title must be revised for the scientific soundness. It is not ML "techniques". Please use "methods". Please note this throughout the paper.
  2. 2. In the abstract, clearly note what ML being used.
  3. 3. please use standard keywords.
  4. 4. The claims in the first paragraph of the abstract not cited. The Research gap is not presented well. The motivation and contribution are not clear in the introduction section. Please answer these questions: why ML methods? why these ML methods? Why not ensemble ML methods? what not hybrid methods? contribution of the paper must satisfy all these questions.
  5. 5. The manuscript is very difficult to follow due to the use of long sentences. Native English proofreading is essential. 
  6. 6. The abbreviations must be explained when first appear. Please insert an Acronyms table.
  7. 7. Please elaborate more on the data and the figure one.
  8. 8. The methods used are not cited. The evaluation metrics are not cited. The former models with these ML methods are not cited.
  9. 9. The result of the paper must be briefly reported in the abstract, and conclusions. Please elaborate on table 2.
  10. 10. What are the suggested ML methods for future research, Please elaborate, and advice readers. 
  11. 11. In respect to the results and the methods used what are the future research direction?
  12. 12. figure 10 must be presented in better quality.

Author Response

This paper presents a comparative analysis of statistical and machine learning methods for probabilistic seismic performance assessment. The paper has merit however many revisions are essential.

Author’s Reply:

First of all, we appreciate the reviewer for taking the time to review our manuscript. Also, we appreciate your positive assessment of our paper. We revised the original manuscript based on the reviewer comments described below. Such revision works considerably helped to improve the quality of the manuscript. Finally, to further improve the quality of the manuscript, the entire sentence was reviewed and further proofread based on such a review.

  1. The title must be revised for the scientific soundness. It is not ML "techniques". Please use "methods". Please note this throughout the paper.

Author’s Reply:

Thank you for the reviewer's suggestion. Concerning the term of machine learning, we have changed the word “techniques” to “methods” in the title and the manuscript according to the reviewer's suggestion. Please, refer to the revised manuscript for the details.

  1. In the abstract, clearly note what ML being used.

Author’s Reply:

Following the reviewer's suggestion, we described what ML methods were used for the abstract. Also, we had the summarized results utilizing such methods be included in the abstract.

  1. please use standard keywords.

Author’s Reply:

Based on the reviewer's comment, the keywords were checked and were corrected in the form of standard keywords. For the details, please refer to the revised manuscript.

  1. The claims in the first paragraph of the abstract not cited. The Research gap is not presented well. The motivation and contribution are not clear in the introduction section. Please answer these questions: why ML methods? why these ML methods? Why not ensemble ML methods? what not hybrid methods? contribution of the paper must satisfy all these questions.

Author’s Reply:

Based on the reviewer's comments, the original manuscript was revised as follows.

  • In the first paragraph of the abstract, we cited the necessary references.
  • In the introduction section, we presented the research gap, the motivation of this study and the contribution of this study.
  • Such a description regarding the research gap, the motivation and the contribution of this study included why machine learning (ML) methods are needed for this study, why we did not use ensemble ML methods and hybrid ML methods.

For the detailed information, please refer to the added and modified contents in the abstract and introduction of the revised manuscript.

  1. The manuscript is very difficult to follow due to the use of long sentences. Native English proofreading is essential.

Author’s Reply:

First of all, we, authors, are sorry for causing such a problem. Accordingly, the long sentences in the manuscript were divided as short as possible, and the native English proofreading was performed on the revised manuscript. Please, refer to the revised manuscript for the details.

  1. The abbreviations must be explained when first appear. Please insert an Acronyms table.

Author’s Reply:

Following the reviewer's comments, we have the abbreviations explained when they first appeared. Also, an acronyms table has been inserted. Please, refer to the revised manuscript for more information.

  1. Please elaborate more on the data and the figure one.

Author’s Reply:

As the reviewer pointed out, the data used in this study were described in more detail. In addition, some modifications have been made for a better understanding of readers with respect to the mentioned Figure one, and Figure one has been described in more detail. For the details, please refer to subsection 2.1 and 2.2.

  1. The methods used are not cited. The evaluation metrics are not cited. The former models with these ML methods are not cited.

Author’s Reply:

Based on the reviewer's comments, the original manuscript was revised as follows:

  • We cited the appropriate references for the methods used in this study.
  • We cited a reference for the evaluation metrics used in this study.
  • The former model for the original data was made only by the traditional statistical methods, not by ML methods. Thus, such an aspect was described and the proper references were cited.

For the detailed revision, please refer to each method description of Subsection 2.3, the explanation of the evaluation metrics of Section 3, and the explanation of the previous model of Section 1 and Subsection 2.2.

  1. The result of the paper must be briefly reported in the abstract, and conclusions. Please elaborate on table 2.

Author’s Reply:

As the reviewer pointed out, we briefly described the results of the paper in the Abstract, within a scope of the Journal’s policy on the length. Also, we reported such results in the Conclusions section, and we more explained about Table 2. For the details, please refer to the related contents in the Abstract, Section 3 and Section 4.

  1. What are the suggested ML methods for future research, Please elaborate, and advice readers.

Author’s Reply:

As the reviewer pointed out, we described the future research in the Conclusions section. Such a description included the suggested ML methods.

  1. In respect to the results and the methods used what are the future research direction?

Author’s Reply:

As the reviewer pointed out, in the Conclusions section, we explained the future research direction with respect to the results and the methods.

  1. figure 10 must be presented in better quality.

Author’s Reply:

We improved the quality of Figure 10 according to the reviewer comment. Please, for the detailed modifications, refer to Figure 10 of the revised manuscript.

Round 2

Reviewer 2 Report

Authors signifcantly improved their manuscript.

Reviewer 4 Report

Dear Authors,

Many thanks for providing the revised version.